# Maintaining Momentum for Rotavirus Immunization in Africa during the COVID-19 Era: Report of the 13th African Rotavirus Symposium

**DOI:** 10.3390/vaccines10091463

**Published:** 2022-09-03

**Authors:** Frederick N. Were, Khuzwayo C. Jere, George E. Armah, M. Jeffrey Mphahlele, Jason M. Mwenda, A. Duncan Steele

**Affiliations:** 1Department of Paediatrics and Child Health, University of Nairobi, Nairobi 00625, Kenya; 2Kenya Paediatric Association, Nairobi 00100, Kenya; 3Malawi-Liverpool-Wellcome Trust Clinical Research Program, Kamuzu University of Health Sciences, Blantyre 312225, Malawi; 4Institute of Infection, Veterinary and Ecological Sciences, University of Liverpool, Liverpool L69 7BE, UK; 5Noguchi Memorial Institute of Medical Research, University of Ghana, Legon, Accra LG 581, Ghana; 6North-West University, Potchefstroom Campus, Potchefstroom 2520, South Africa; 7WHO Regional Office for Africa, Brazzaville P.O. Box 2465, Congo; 8Department of Virology, Sefako Makgatho Health Sciences University, Pretoria 0204, South Africa

**Keywords:** African rotavirus symposium, diarrhea, rotavirus, rotavirus vaccine, SARS-CoV2, COVID-19 pandemic

## Abstract

The 13th African Rotavirus Symposium was held as a virtual event hosted by the University of Nairobi, Kenya and The Kenya Paediatric Association on 3rd and 4th November 2021. This biennial event organized under the auspices of the African Rotavirus Network shapes the agenda for rotavirus research and prevention on the continent, attracting key international and regional opinion leaders, researchers, and public health scientists. The African Rotavirus Network is a regional network of institutions initially established in 1999, and now encompassing much of the diarrheal disease and rotavirus related research in Africa, in collaboration with the World Health Organization African Regional Office (WHO-AFRO), Ministries of Health, and other partners. Surges in SARS-CoV2 variants and concomitant travel restrictions limited the meeting to a webinar platform with invited scientific presentations and scientific presentations from selected abstracts. The scientific program covered updates on burden of diarrheal diseases including rotavirus, the genomic characterization of rotavirus strains pre- and post-rotavirus vaccine introduction, and data from clinical evaluation of new rotavirus vaccines in Africa. Finally, 42 of the 54 African countries have fully introduced rotavirus vaccination at the time of the meeting, including the two recently WHO pre-qualified vaccines from India. Nonetheless, the full benefit of rotavirus vaccination is yet to be realized in Africa where approximately 80% of the global burden of rotavirus mortality exists.

## 1. Introduction

Diarrheal diseases are a significant cause of mortality in young children under 5 years of age where 1 in 10 childhood deaths are due to diarrheal disease, resulting in more than half a million childhood deaths every year [1]. The highest risk remains evident in sub-Saharan Africa, where the reduction in diarrheal incidence over time has not been as consistent as that seen in other regions such as South Asia and Latin America. Furthermore, disparities in known interventions to protect children or prevent infection remain disproportionately high in Africa. In a recent report, oral rehydration therapy (ORT) was documented as unavailable or inadequately utilized in several countries in Africa, including the Democratic Republic of Congo (DRC) and Nigeria, two countries with large birth cohorts and high rotavirus mortality [2]. Certainly, the distribution and use of ORT is very variable in many countries in Africa. In addition, although access to safe water and improved sanitation facilities (WaSH) have increased globally over time between 2000 and 2017, sub-Saharan Africa showed fewer overall improvements in access and in some sub-national areas, lack of safe drinking water and improved sanitation facilities remained high [3]. It is no coincidence that considerable overlap exists between sub-Saharan countries without substantial improvements in ORT coverage, WaSH interventions, and improved nutrition, and a significant proportion of diarrheal mortality burden. 

Among the more than 500,000 infants and young children who perish every year from an acute diarrheal episode, rotavirus remains the most predominant pathogen and is associated with >200,000 of those deaths [4]. It is not surprising, therefore, that the WHO has strongly recommended rotavirus immunization as a key intervention to control rotavirus diarrhea and to prevent childhood deaths from rotavirus infection. Rotavirus immunization in Africa was proceeding well with 42 of the 54 countries fully implementing rotavirus immunization at the time of the last meeting in November 2019 [5]. Africa contributes approximately 80% of the global burden of rotavirus mortality [4], so the scale up of rotavirus vaccination across the continent was significant. In addition, although several countries had documented the impact of routine immunization on reductions in diarrheal deaths and hospitalizations post-rotavirus vaccine introduction [6,7,8], the desired full benefit is yet to be realized. In December 2019, the DRC and Benin introduced a rotavirus vaccine and Nigeria was poised to introduce in 2020 before the COVID-19 pandemic arrived. Nigeria and DRC have some of the highest rates of rotavirus mortality per 1000 live births and rotavirus immunization was anticipated to achieve a major impact on the global figures for mortality with their introduction of vaccine [4]. 

The WHO has pre-qualified four rotavirus vaccines, making them eligible for vaccine subsidy support through Gavi, The Vaccine Alliance; variations of three of these products are currently available (Table 1). Rotarix, (GSK Biologicals, Rixensart, Belgium) is a monovalent human rotavirus strain that demonstrated efficacy in Africa, was WHO pre-qualified in 2009, and is the most widely used product on the continent. Several studies have presented robust effectiveness against rotavirus diarrhea and death [9]. Rotavac vaccine (Bharat Biotech, Hyderbad, India) derived from a novel, naturally reassorted human-bovine monovalent strain that was identified in India showed safety and efficacy in India and was prequalified in 2018. The vaccine is available in at least two formulations with different temperature characteristics and vaccine vial monitors (VVM)—VVM-2 for Rotavac or VVM-7 for Rotavac 5D (Table 1). The vaccine has been introduced in several countries in Africa. RotaSIIL (Serum Institute, Pune, India) is a bovine-human reassortant vaccine developed at the US NIH, carrying the five most common outer protein VP7 antigens of human rotaviruses. The vaccine demonstrated efficacy in clinical trials in India and Niger, was pre-qualified in 2018, and exists in three formulations with either a lyophilized or liquid formulation. These products are illustrated in Table 1 (derived and modified from ref. [10]). RotaTeq (Merck, White River, PA, USA) is the original pentavalent bovine-human reassortant rotavirus vaccine and is widely used globally in upper-middle and high-income countries. 

The 13th African Rotavirus Symposium was held as a virtual event on 3rd and 4th November 2021 during the upsurge of the Omicron strain of the COVID-19 pandemic. During the opening of this symposium hosted in Nairobi, Kenya, the Acting Director General of Health in the Ministry of Health, Dr. Patrick Amoth, expressed his disappointment that Kenya—one of the first countries in Africa to introduce rotavirus vaccines in 2012—was unable to host the Symposium in-person due to the COVID-19 pandemic. Dr. Amoth highlighted the impact that the introduction of rotavirus vaccines has had in Kenya leading to reductions in approximately 67% of rotavirus hospitalizations. Rotavirus immunization was estimated to prevent over 20,000 deaths in 2016 alone in those countries in Africa using the vaccine, and this could be dramatically increased if all countries introduced the vaccine. Dr. Amoth also recognized the burden that still exists globally with >200,000 rotavirus-associated deaths, most in Africa, and he encouraged African countries to introduce the vaccine where it was not being used yet. In addition, Dr. Amoth emphasized the significance of two recently pre-qualified vaccines available to Gavi (i.e., Rotavac and RotaSIIL) and, as some countries switch to these products, the need for studies in Africa to assess the impact of the new vaccines, their cost-effectiveness, and programmatic implications.

## 2. Rotavirus Immunization in Africa and the Impact of COVID-19

Dr. Jason Mwenda, WHO Regional Office for Africa (WHO-AFRO) described the progress in rotavirus vaccine implementation across the continent and the challenges that have arisen during the global COVID-19 pandemic. The pandemic has not spared the continent, with cumulative cases topping 6 million during the period March 2020 and October 2021 and a case fatality rate of 2.5% resulting in >150,000 deaths in all ages [11]. Recognizing that robust surveillance and diagnostic tools have not been widely available in Africa, these numbers are highly likely under-reported. Three classic epidemic waves of SARS-CoV-2 variants have been observed over this time period. To combat the pandemic, as of October 2021, >250,000,000 doses of COVID-19 vaccine had been procured through various sources including bilateral agreements between countries and vaccine manufacturers, the COVAX facility—an initiative co-led by Gavi, the Coalition for Epidemic Preparedness Innovations (CEPI) and the WHO—or African Union-led efforts through the Africa Vaccine Acquisition Task Team (AVATT) for 52 countries in Africa. Approximately 68% of the vaccines have been administered (~172,000,000 doses) as of late October 2021. Nevertheless, only 73 million people have been fully immunized representing 5.4% of the population on the continent. As of October 2021, Africa represented only 3% of the doses administered globally [11]. 

The pandemic has impacted many public health programs in Africa including the introduction of rotavirus vaccines, surveillance for vaccine-preventable diseases, and other health services. Since the last African Rotavirus Symposium in 2019 [12], Benin and DRC introduced rotavirus vaccine in late 2019. Unfortunately, planned introductions in Central African Republic and Nigeria were delayed by the pandemic. In addition, several countries have conducted switches of their rotavirus products driven by global vaccine supply issues. The withdrawal of the RotaTeq vaccine from the Gavi market in 2018 led to several countries switching vaccine products in 2020—Burkina Faso and Mali switched from RotaTeq to RotaSIIL in early 2020, Cote d’Ivoire switched to Rotarix, and Ghana switched from Rotarix to Rotavac in January 2020. Further switches are anticipated in 2022/2023. 

Unfortunately, the pandemic did not only delay some rotavirus vaccine introductions but the significant gains in routine vaccine coverage since the advent of Gavi in 2000 were impacted overall. Coverage of a third dose of vaccine protecting against diphtheria, tetanus, and pertussis (DTP-3) dropped to 83% in 2020, making 22.7 million children vulnerable to these vaccine-preventable diseases [13]. Dr. Antoinette Ba-Nguz, Immunization Regional Coordinator for UNICEF, addressed the impact of the COVID-19 pandemic on the hard-fought gains in routine immunization in the African region. A WHO interim report on pulse surveys evaluating the impact of the COVID-19 pandemic on essential health services was conducted in over 100 countries, including 30 from Africa. On average, countries reported significant disruptions to more than 50% of health services, with routine immunization services, family planning, and ante-natal care the hardest hit [14]. Dr. Ba-Nguz also described modelling data that suggested that the indirect effects of COVID-19 disruptions to the health services were likely to exceed that of the direct effect of COVID-19 infections. Without focused efforts to make up the missed immunization visits for children, we will see residual impact of the pandemic in the years ahead. 

Dr. Latif Ndeketa, Malawi-Liverpool-Wellcome Trust Clinical Research Programme, outlined the rollout of COVID-19 vaccines in Africa. He presented similar data to that from UNICEF and AFRO highlighting the inequity of COVID-19 vaccine introduction in the region. However, vaccine shipments saw a sharp increase in the delivery of doses to countries from mid-August 2021, thanks in large part to the COVAX partnership. Nevertheless, most countries have struggled to keep up with immunization campaigns to administer the vaccine doses received and most are struggling to meet the WHO Director’s general recommendation to reach 10% of the population with at least 1 dose [11]. Interestingly, the distribution of vaccines in Africa at the end of October 2021 showed the AstraZeneca/CoviShield vaccine comprised almost a third of all vaccine doses utilized (31%), followed by Sinopharm (20%) and Sinovac (18%) from Chinese manufacturers. The Pfizer and J&J vaccines were listed as 13% and 11% respectively while Moderna (5%) and Sputnik (2%) completed the available repertoire. 

Finally, Dr. Frederick Debullet, PATH described an updated cost effectiveness analysis of rotavirus vaccines given the new products in the Gavi market and the upheavals created by the pandemic. He noted that rotavirus vaccines avert a substantial rotavirus disease burden in Africa and that despite financing and pandemic challenges, in all but one country, rotavirus vaccines were a cost-effective intervention [15]. Given the wide array of rotavirus vaccine products now available, he encouraged countries to examine which specific product best suited their specific health, programmatic, and cost needs. 

## 3. Programmatic Use of Rotavirus Vaccines in Africa

Dr. Jackie Tate, US CDC, described results of several efforts to elucidate the occurrence and possible association of rotavirus immunization and intussusception. It is the most common cause of bowel obstruction in infants and occurs naturally when one segment of the bowel folds into another, causing a blockage. An early generation rotavirus vaccine was associated with intussusception in post-licensure evaluations following its introduction into the national immunization program in the United States in the late 1990s. For second generation rotavirus vaccines which became available in 2006, no increased risk of intussusception was identified in large clinical trials for Rotarix or RotaTeq [16]. Post-licensure evaluations identified a small increased risk in some high- and middle-income countries for Rotarix and RotaTeq of 1–6 excess cases per every 100,000 vaccinated infants [16]. However, no increased risk has been identified in post-licensure evaluations in low and middle-income countries in Africa with Rotarix [17]. The objective of this new evaluation was to monitor the safety of rotavirus vaccines under routine public health use among infants in African countries introducing pentavalent rotavirus vaccine (RotaTeq) and specifically to assess any potential association between RotaTeq vaccination and intussusception after national introduction of the vaccine in the routine childhood immunization schedules.

The African Intussusception Surveillance Network was formally established in 2014 and all countries in the network used a common protocol for comparability across sites/countries that allowed for pooling of data for analysis. Enrollment in countries using RotaTeq was completed in 2021. Active surveillance for cases of intussusception meeting the Brighton Collaboration case definition for level 1 of diagnostic certainty in children <12 months of age was conducted at 15 sentinel pediatric hospitals in five countries (Burkina Faso, Cote d’Ivoire, The Gambia, Mali, and Rwanda). Vaccination status for enrolled cases was confirmed either by the child’s vaccination card or the vaccine registry maintained by the vaccination clinic. A self-controlled case-series analysis was conducted to determine whether an association exists between rotavirus vaccination and intussusception. A slightly elevated but non-significant risk of intussusception was observed in the 1–7 days following dose 1 of RotaTeq. However, the vaccine was administered at a time when rates of intussusception were naturally low, so the absolute number of cases was small. RotaTeq is no longer used in Gavi-eligible countries; however, these findings highlight the need for vaccine product-specific risk monitoring, particularly as countries switch to, or introduce, the two new rotavirus vaccines. These studies are planned with some of the new country adoptions.

Benin introduced Rotavac in December 2019 after its WHO pre-qualification in 2018, reported Dr. Aristide Sossou, Ministry of Health (MOH). Benin has conducted hospital-based diarrhea surveillance within the African Rotavirus Surveillance network since 2013 and between 30% and 40% of diarrhea cases were positive for rotavirus antigen by ELISA. The 5-dose vial of rotavirus was implemented in the same schedule as the national schedule of pentavalent vaccine (DPT-HepB-HiB) at 6, 10, and 14 weeks of age. Despite interference from the pandemic, which hampered initial rotavirus vaccine coverage rates, by January 2021 high rates of Rotavac coverage were achieved reflecting routine EPI rates in the country.

Dr. Andre Tonda, Ministry of Health and Family Welfare (MOHFW) presented the introduction of the lyophilized RotaSIIL vaccine into the childhood immunization program in DRC. Rotavirus surveillance has been ongoing in DRC since the end of 2009 demonstrating the high burden of diarrheal disease in the birth cohort of almost 4.5 million infants. In September 2016, the MOHFW submitted a Gavi application for the introduction of the 2-dose Rotarix, triggering a review of the readiness of the country with respect to the cold chain capacity, internal transport and logistics and micro-planning for the introduction. After national review and preparation based on these analyses for 26 provinces, a progressive introduction starting in 3 phases was proposed. Global supply issues meant a change in product, and RotaSIIL, pre-qualified in 2018, was introduced at the end of 2019. By the end of August 2021, third rotavirus vaccine dose coverage was at 76.2%, almost 10% lower than that seen with pentavalent or pneumococcal conjugate vaccines.

Dr. Kwame Amponsa-Achiano, EPI Manager, Ministry of Health in Ghana discussed the challenges associated with the switch of products from the 2-dose Rotarix to a 3-dose Rotavac in the birth cohort of over 1.2 million infants. Ghana introduced Rotarix in May 2012 based on decades of rotavirus surveillance which documented high disease burden in 40%–50% of cases of diarrhoea attending hospitals. In 2020, Ghana MOH applied for a switch of rotavirus vaccine product to Rotavac based on multiple considerations including the costs of the vaccine (Ghana considered their trajectory transitioning from full Gavi subsidy support), the cold chain footprint, and the efficacy and safety database of the vaccine in India. Careful micro-planning was initiated to review the cold chain capacity required, use of available Rotarix vaccines at clinics, revised training materials and the immunization records for children and the communications with both health care workers and families [18]. This was noted to facilitate a smooth transition.

## 4. Diarrhea and Rotavirus Surveillance in Africa

Dr. James Platts-Mills, University of Virginia, USA, provided estimates of the etiology-specific burden of hospitalized diarrhea cases from the WHO Global Pediatric Diarrhea Surveillance Network (GPDS), which enrolled almost 30,000 <5-year-old children presenting with bloody and non-bloody diarrhea of any duration in 33 countries from Africa, Asia, Latin America and Europe in 2017 and 2018. Dr. Platts-Mills focused on data generated from 7880 children in 10 African countries: four from West Africa (Benin, Ghana, Cote d’Ivoire and Nigeria), and the remainder from Eastern and Southern Africa (Ethiopia, Uganda, Zambia, Zimbabwe, Mauritius and Madagascar). Despite the introduction of rotavirus vaccines in most African countries, a previous report documented that rotavirus remains the most prevalent pathogen associated with diarrhoea requiring hospitalization in African children [4]. Rotavirus was the leading cause of hospitalized diarrhea in seven countries. Exceptions were the sites in Cote d’Ivoire and Ethiopia where *Shigella* was the leading cause, and Mauritius where norovirus was the leading cause. 

They then used estimates of the incidence of hospitalized diarrhea in each country to aggregate the site level estimates into geographic grouping. In GPDS, as a whole, rotavirus was associated with the most diarrhea hospitalizations, followed by *Shigella*, norovirus, and enteric adenoviruses (40/41). Countries that had introduced rotavirus vaccines had a lower burden of rotavirus (AF 21.3; 95% CI 18.1, 25.0) compared with countries that had not (AF 48.3; 95% CI 34.4, 65.5) (Platts-Mills et al., manuscript submitted for publication). 

Dr. Platt-Mills used the etiology of hospitalized diarrhea in GPDS as a proxy to estimate causes of diarrheal deaths in 2017 and 2018, incorporating estimates of country-specific diarrheal deaths from the Global Burden of Disease (GBD) study. The total number of all-cause diarrheal deaths was estimated at 582,292 based on the GBD 2020 models. Of these, 396,459 deaths occurred in the African region, of which, rotavirus (148,931), *Shigella* (43,947), norovirus (19,562), enterotoxigenic *E. coli* (18,879), and *Cryptosporidium* spp. (17,121) were the top five attributing pathogens. Given the impact of rotavirus vaccines in reducing hospitalization and death noted above [6,7], improving the efficacy and coverage of rotavirus vaccination and prioritizing interventions against other enteric pathogens including *Shigella* and norovirus could further reduce diarrhea morbidity and mortality.

Dr. Karen Kotloff, University of Maryland School of Medicine, USA, presented data from the Rotavirus Vaccine Impact on Diarrhea in Africa (RVIDA) study where she demonstrated the changing landscape of moderate-to-severe diarrhea (MSD) among children in Kenya, Mali, and The Gambia following rotavirus vaccine introduction. Like the Global Enteric Multicenter study (GEMS) [19], RVIDA characterized the etiology associated with diarrhea episodes using conventional microbiology and quantitative polymerase chain reaction (qPCR) methods to determine pathogen-specific attributable fractions and pathogen-attributable incidence in stool samples. The impact of MSD was assessed on mortality and linear growth faltering through health care utilization and vaccine coverage surveys [20]. As both GEMS and RVIDA were population-based studies with age-matched controls that used similar standardized clinical, epidemiological, and microbiology methods, comparison of the results from these two studies enabled calculation of pathogen-specific incidence/attributable fraction of diarrhea. 

Rotavirus and *Cryptosporidium* spp. were significantly associated with MSD in all three countries in all age groups although infections peaked during infancy (0–11 months). The attributable incidence of rotavirus was the highest among diarrhea pathogens associated with MSD in children under 12 months (3.1 episodes per 100 child-years, 95% CI 2.7–3.5), while the incidence of *Shigella* was significantly higher than other pathogens in children between 12–59 months of age with peak infections occurring in the second year of life (5.4 episodes per 100 child-years, 95% CI 4.9–6.4). The researchers found several epidemiological changes, for instance (i) rotavirus diarrhea incidence declined by ~54% post rotavirus vaccine introduction, (ii) the burden of *Shigella* and *Cryptosporidium* across all age groups was significant, and (iii) the viral agents collectively (i.e., enteric adenoviruses 40/41, noroviruses, sapoviruses, and astroviruses) contributed a large disease burden. Finally, an episode of MSD continues to be associated with linear growth faltering and mortality.

Prof. Beckie Tagbo, University of Nigeria, presented rotavirus surveillance data from Southeast Nigeria. They assessed the general trend of rotavirus diarrhea cases over a 10-year period (2011–2021) and examined the possible effects of the delay in introduction of rotavirus vaccine in Nigeria. They also assessed the possible impact of the COVID-19 pandemic on the rotavirus surveillance platform. Rotavirus-associated diarrhea was high, and the burden of rotavirus disease was on the increase among children <24 months attending hospitals. The enrolment significantly dropped in 2021 by 86.6% which was attributed to the COVID-19 pandemic surge. She called for action from the Nigerian Government and local and international partners to hasten the introduction of rotavirus vaccines to avert further diarrhea-associated mortality and morbidity.

## 5. Genomic Analysis of Rotavirus Strains in Africa 

Dr. Khuzwayo Jere, University of Liverpool and the Malawi-Liverpool-Wellcome Trust Clinical Research Programme presented an overview of rotavirus strains that have emerged after introduction of rotavirus vaccine in Malawi. Several studies have documented homotypic and heterotypic immune responses and protection from both the monovalent and pentavalent vaccines [21], although this question remains a colorful topic of discussion in the rotavirus community. Data from Malawi showed that monovalent Rotarix was more effective against homotypic strains compared with heterotypic strains [7], and the genetic backbone of the strain was not a significant driver of vaccine effectiveness but rather the two outer capsid proteins, the VP7 and VP4 genotypes, played this role [22]. Long-term hospital-based diarrhea surveillance spanning 22 years in Blantyre demonstrated the rich diversity of rotavirus strains circulating in children hospitalized with diarrhea [23]. Genotypes G1P[8] and G2P[4] predominated prior to vaccine introduction in 2011, declining in each successive year until 2018, after which they were not detected. In contrast, rotavirus strains that prevailed in the 1990s re-emerged post-2018, notably G3, which became the predominant genotype by the end of 2019. Whole genome sequencing (WGS) of the re-emergent G3 strains identified them as all ‘typical’ human G3 strains with either P[8] or P[6] VP4 genotypes, unlike the reassortant equine-like G3s that have been seen in other countries [24,25]. Interestingly, the Malawian G3 strains fell into two groups—those with the VP4 P[4] and P[6] genotype had a DS-1-like genetic constellation (genogroup 2), whereas those associated with a P[8] genotype had a Wa-like genetic constellation (genogroup 1). 

Data on the impact of rotavirus vaccine on the trending diversity of rotavirus strains in Ghana were presented by Dr. Francis Dennis, University of Ghana, based on their long-term rotavirus surveillance. The story is reassuringly similar to that in Malawi. Wide rotavirus strain diversity was reported before vaccine introduction (2006–2012) with G1P[8] strains the most predominant in most years. G2 strains were detected at higher frequencies in certain years mirroring the picture in pre-vaccine era in Malawi. Following vaccine introduction (2013–2019), novel strains emerged and predominated. For instance, G12 strains with VP4 P[6] or P[8] were common in 2013–2014 and G9P[6] strains were frequently detected between 2015 and 2018 [26]. WGS analysis of pre-vaccine strains showed that G1 rotaviruses had a Wa-like genetic backbone (genogroup 1) while G2, G3, and G9 strains had a DS-1-like genetic backbone (genogroup 2). Similar profiles were observed for all post-vaccine strains with the exception of G9 strains exhibiting a DS-1-like genetic constellation with unusual reassortant E6 NSP4 genotype. Dr. Dennis concluded by showing that the COVID-19 pandemic has affected their diarrhea surveillance in Ghana where cases were significantly lower during the COVID-19 era compared with the years before in the Navrongo Demographic Health Service site. Fortunately, rotavirus vaccine coverage has not been significantly impacted as anticipated, with coverage at 89% post-COVID-19 compared to 90% pre-pandemic.

Ethiopia presented a slightly different picture. Dr. Mesfin Tefera, Ethiopian Public Health Institute, Ethiopia, presented data on rotavirus strain diversity in Ethiopia from 2011–2018 spanning the introduction in 2014. G2P[4] and G12P[8] strains were predominant before Rotarix vaccine introduction (2011–2013), whereas G3P[8] and G3P[6] rotaviruses predominated during the vaccine era (2014–2018). Overall, the data presented in these studies exhibited that the diversity and distribution of rotavirus strains are not significantly driven by vaccine pressure but do reflect the well-recognized process of natural genotype oscillation driven by antigenic drift and antigenic shift of the segmented RNA virus.

Dr. Peter Mwangi, University of the Free State, South Africa presented findings from the African Enteric Viruses Genome Initiative (AEVGI). They analyzed the whole genome sequences of the South African G1P[8] and G2P[4] rotavirus strains over a 14-year period [27]. With the exception of one atypical pre-vaccine G1P[8] strain that had a DS-1-like constellation, the rest of the G1P[8] and G2P[4] strains were “typical” as they fell into the two classic genogroups, Wa-like or DS-1-like, respectively. Rotarix introduction did not impact evolutionary changes on G1P[8] strains. The atypical pre-vaccine strain was generated through natural reassortment events. Two distinct pre- and post-vaccine G2P[4] sub-lineages were identified which likely resulted from step-wise sub-lineage evolution of G2P[4] strains. Lineage-defining amino acid sequences were identified in each gene. They also found that the stabilizing effect of the amino acid mutation on the outer capsid protein structure was a likely means by which the G2P[4] viruses enhanced their fitness over time, while purifying selection was utilized to purge disadvantageous random mutations among the viral population. Vaccine-selective pressure on G2P[4] strains could not be ruled out though, and hence further longitudinal whole-genome sequencing studies to assess the full impact of vaccine use on circulating rotaviruses are essential.

Dr. Wairimu Maringa, University of the Free State, South Africa, presented whole-genome sequencing-based rotavirus surveillance data from Zambia. Although all strains that were sequenced had either a Wa-like (genogroup 1) or DS-1- (genogroup 2) genetic constellation, their data revealed substantial diversity of rotavirus strains that circulated before and after introduction of Rotarix vaccine in 2014. Interestingly, novel strains—G8P[4], G9P[6], and G12P[6]—were only detected before vaccine introduction, whereas several G1 and G2 reassortant strains were identified only during the vaccine era. An unusual G5P[6] strain was fully characterized from a Zambian child—most of its genome segments resembled those of animal strains, and it was hence thought to have arisen through zoonotic transmission coupled with reassortment events [28]. Various amino acid variations were observed when the strains from Zambia were compared to that of Rotarix strains. Whether such strains with animal rotavirus-like characteristics, or those with various amino acid changes on their outer capsid proteins, would have a negative impact on the vaccine effectiveness remains to be determined.

## 6. Rotavirus Vaccine Research in Africa 

African researchers continue to lead the way with respect to the clinical evaluation of new and novel rotavirus vaccines. Dr. Roma Chilengi, Centre for Infectious Disease Research (CIDRZ), Lusaka, Zambia described an immunogenicity study of the Rotavac vaccine. Rotavac is available in two formulations—a frozen formulation stored at −20 °C with 2–6 months stability at 2–8 °C (VVM-2) and Rotavac 5D, which is a liquid formulation of the vaccine stable at 2–8 °C for 24 months (VVM-7). This study evaluated and compared the safety and immunogenicity of both formulations in an infant population in Zambia in a Phase 2b, open label, randomized, controlled trial; 450 infants 6 to 8 weeks of age were randomized equally to receive three doses of Rotavac or Rotavac 5D, or two doses of Rotarix. Study vaccines were administered concomitantly with routine immunizations. Blood samples were collected pre-vaccination and 28 days after the last dose. Serum anti-rotavirus IgA antibodies were measured by ELISA, using viral lysates from rotavirus strains WC3 (GxP[5]) and 89–12 (G1P[8]) in the assays which were obtained from Cincinnati Children’s Hospital and Medical Centre (CCHMC). The primary analysis was to assess non-inferiority of Rotavac 5D to Rotavac in terms of the geometric mean concentration (GMC) of serum IgA (WC3) antibodies. Sero-response and seropositivity were also determined. Safety was evaluated as occurrence of immediate, solicited, unsolicited, and serious adverse events after each dose [29]. 

The study evaluated 388 infants in the per-protocol population. All three vaccines were well tolerated and immunogenic. The post-vaccination GMCs were 14.0 U/mL (95% CI: 10.4, 18.8) and 18.1 U/mL (95% CI: 13.7, 24.0) for the Rotavac and Rotavac 5D groups, respectively, yielding a ratio of 1.3 (95% CI: 0.9, 1.9), thus meeting the pre-set non-inferiority criteria [29]. Solicited and unsolicited adverse events were similar across all study arms. No death or intussusception case was reported during the study period. Among Zambian infants, both Rotavac and Rotavac 5D were well tolerated and the immunogenicity of Rotavac 5D was non-inferior to that of Rotavac. These results are consistent with those observed in licensure trials in India and support use of these vaccines across wider geographical areas.

Dr. Julie Bines, Murdoch Children’s Research Institute at the Royal Children’s Hospital in Melbourne reported data from a dose-ranging study of the neonatal RV3-BB vaccine in Malawian infants at three primary health clinics in Blantyre. This Phase 2, randomized, double-blind parallel study evaluated three doses of RV3-BB in healthy neonates less than 6 days of age with a birthweight between 2.5–4.0 kg. Subjects were randomized in a 1:1:1:1 strategy to receive either high-titer (1.0 × 10^7^ focus-forming units (FFU) per mL), mid titer (3.0 × 10^6^ FFU per mL), or low titer (1.0 × 10^6^ FFU per mL) in the neonatal cohorts or an infant group which received the high-titer dose. Neonates received the three-dose schedule at 0–5 days and 6 and 10 weeks of age while the infant group received the doses at 6, 10, and 14 weeks of age [30]. 

In the neonatal schedule, cumulative IgA seroconversion 4 weeks after three doses of RV3-BB (i.e., at 14 weeks of age) was observed in 79 (57%) of 139 participants in the high-titer group, 80 (57%) of 141 participants in the mid-titer group, and 57 (41%) of 138 participants in the low-titer group. No difference in cumulative IgA seroconversion 4 weeks after three doses of RV3-BB was observed between high-titer and mid-titer groups in the neonatal schedule (difference in response rate 0.001 (95% CI: −0.115 to 0.117)), fulfilling the criteria for non-inferiority. In the infant schedule group, 82 (59%) of 139 participants had a cumulative IgA seroconversion 4 weeks after three doses of RV3-BB (i.e., at 18 weeks of age) [30]. Three doses of RV3-BB were well tolerated with no difference in adverse events among treatment groups. RV3-BB was well tolerated and immunogenic when co-administered with routine childhood EPI vaccines in a neonatal or infant schedule. A lower titer (mid-titer) vaccine generated similar IgA seroconversion to the high-titer vaccine, presenting an opportunity to enhance manufacturing capacity and reduce costs. Neonatal administration of the RV3-BB vaccine has the potential to improve protection against rotavirus disease in children in a high-child mortality country in Africa.

## 7. Conclusions

Despite the effect of the global COVID-19 pandemic on rotavirus surveillance and, more importantly, on rotavirus vaccine introductions and impact on diarrheal disease across the continent, we are seeing some rebound after these interruptions. However, there is considerable work to be done to strengthen the routine rotavirus vaccine coverage across the continent, to address inequities in many countries with respect to ORT and WaSH interventions, and to improve access to health care for all children. 

African researchers continue to lead the international agenda with respect to the evaluation of new rotavirus vaccine candidates including the recently licensed and pre-qualified vaccines, Rotavac, Bharat Biotech and, RotaSIIL, Serum Institute; a novel neonatal strain, RV3-BB developed by Australian researchers with PT biofarma, Bandung, Indonesia; and the next generation parenteral rotavirus vaccine [31], which is undergoing a multi-country phase 3 safety and efficacy study currently.

## Figures and Tables

**Table 1 vaccines-10-01463-t001:** WHO pre-qualified oral, live-attenuated rotavirus vaccines available for country implementation (modified from reference [10]).

**Manufacturer**	GSK	Bharat Biotech		Serum Institute		
**Trade Name**	Rotarix	Rotavac	Rotavac 5D	RotaSIIL	RotaSIIL liquid	RotaSIIL thermo
**NRA ^a^**	Belgium	India		India		
**Form**	Liquid	Liquid frozen	Liquid	Lyophilised	Liquid	Lyophilised
**Presentation**	Plastic tube	Glass vial	Glass vial	2 glass vials	2 glass vials	2 glass vials
**VVM ^b^ Type**	7	2	7	30	7	250
**Doses/Vial**	1 dose	5 or 10	1 or 5	1 or 2	1 or 2	1 or 2
**WHO PQ ^c^ Date**	2009	2018	2021	2018	2021	2020
**Presentation**	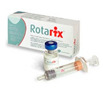	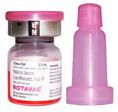 Liquid formulation in 1, 5 or 10 dose vials.		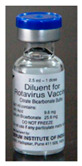	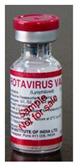	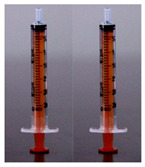
**Comments**		Two products Rotavac—VVM-2 (2–6 months at 2–8 °C) Rotavac 5D—VVM-7 (24 month stability at 2–8 °C)		Three products; two lyophilised RotaSIIL—lyophilized, VVM-30 (30 months at 2–8 °C) RotaSIIL thermo—lyophilized, VVM-250 (30 months at <25 °C) RotaSIIL liquid—VVM-7 (24 months at 2–8 °C)		

^a^ National Regulatory Authority; ^b^ Vaccine Vial Monitor; ^c^ Pre-qualification.

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
