# Peer review of "Maintaining Momentum for Rotavirus Immunization in Africa during the COVID-19 Era: Report of the 13th African Rotavirus Symposium"

_vaccines, 2022, doi:10.3390/vaccines10091463_

Round 1
Reviewer 1 Report
This is a very good summary of a meeting covering all issues of rotavirus in Africa: Achievements related to vaccines, discussion of the various available vaccines, impact of ongoing illness, ideas for future work.
It offers a very good reference for researchers and clinicians working on rotavirus.
Author Response
Thank you for reviewing our manuscript.
The overview of the meeting presentations included in this manuscript were anticipated to provide a broad update on the situation in Africa. We appreciate your comments that it has achieved this objective.
Reviewer 2 Report
This study, in which the authors investigated the effects of the pandemic on the rotavirus vaccination schedule in African countries, was well designed and written. The article contains sufficient information.
Strenghts:
The authors investigated whether the rotavirus vaccine program continued in the African continent during the course of the COVID-19 pandemic. While rotavirus infection is only one cause of diarrhea in the Western world, in the African continent this infection causes the death of many children. Therefore, it is extremely important that rotavirus vaccination continues in Africa. Many healthcare services have been disrupted during the COVID-19 pandemic. The study provides important information about whether rotavirus vaccination in Africa has been disrupted during the pandemic.
Limitations:
The study collected data in the acute phase. The COVID-19 pandemic still continues. Therefore, the study only provides information about the first years of the pandemic. As the health system weakens during the pandemic, there may be doubts about the accuracy of data. It also did not provides precise information about the prevalence of rotavirus infection during the pandemic.
Author Response
Thank you for your review of our manuscript.
As you state - this virtual meeting was held while the pandemic was still active and the surge in Omicron was still happening. The discussion of the impact of the pandemic on rotavirus disease and rotavirus immunization programs is, therefore, limited to the timing of the meeting.
The manuscript describes the existing situation in several African countries and the impact on the rotavirus immunization rates and rotavirus surveillance data during the later stages of the pandemic and during the Omicron surge.
- Recent reports from WHO have confirmed the decline in immunization coverage rates globally (WHO, 2022).
- We do not believe that rotavirus prevalence rates declined per se, but that health care seeking behaviors for parents of ill children did change and they did not attend Health Care Clinics at the same level as previously.
Importantly, we are now planning the 14th African Rotavirus Symposium for 2023 and the Scientific Organizing Committee has emphasized the need to monitor the changes that may have occurred in health-seeking behavior and immunization coverage rates for discussion at that meeting.
I hope that this addresses the concerns you have about the accuracy of the data presented. It was the data available in October 2021 (just prior to the meeting in November 2021).
Reviewer 3 Report
In the present review, authors summarized pre-qualified four rotavirus vaccines, the rotavirus immunization in Africa and the impact of COVID-19, the programmatic use of rotavirus vaccines and rotavirus surveillance in Africa, as well as the rotavirus vaccine research including the genomic analysis of rotavirus strains in Africa. However, there are some minor points need be addressed as following.
1. In the table 1, it is better to describe the abbreviation such as VVM, NRA in the table notes.
2. In addition, please double check the information as follow:
(1) Line 96 in table 1, “2-6 months at 2-8C”, “30 months at 2-8C”, “30 months at <25C”;
(2) Line 329-332: “Whole genome sequencing 329 (WGS) of the re-emergent G3 strains identified them as all ‘typical’ human G3 strains with either P[8] or P[6] VP4 genotypes, unlike the reassortant equine-like G3s that have been seen in other countries [24,25].” The genotype of rotavirus strains is P, P8 or P6?
3. It is suggested to think about what is the specific genetic characterization of rotavirus strain in Africa? Aims to preventing the dominate rotavirus strains in Africa, what is focus in the rotavirus immunization research?
Author Response
Thank you for your review of our manuscript and the suggestions to improve it.
- I have added the abbreviations to Table 1.
- I have checked the information in Table 1 (line 96). It is correct. Please see this further clarification:
- One version of the RotaSIIL product was approved for use out of the traditional cold chain and has a Vaccine Vial Monitor of 30 for the lyophilized vaccine and diluent (VVM30 = stability at 2-8C for 30 months).
- Serum Institute then filed for a thermostable license of the same product which has a VVM of 250; however the buffer diluent needs to be stored at temperatures below 25C, thus leading to the parameters of 30 months stability.
- Their final product is a liquid product which negates requiring two vials (ie. one for the lyophilized vaccine cake, and the second for the 2.5mL diluent buffer per dose). This has a VVM of 7 (24 months stability at 2-8C).
- The section in lines 329-332 is a summary of several research projects identifying and characterizing rotavirus strains in Africa. There were additional reports in poster format supporting this data.
For clarity - rotavirus strains are characterized on the basis of two outer capsid neutralization antigens. These are the VP7 glycoprotein or G-type and the VP4 protease-sensitive antigen or P-type. The most common human rotavirus VP4 P-types are P[8], P[6] and P[4] characterized by gene sequences encoding the type-specific epitopes.
The combination of different G- and P- types are usually dependent on the viral genogroup to which they belong - P[8] and P[6] strains usually belong to Genogroup I, whereas P[4] strains cluster in Genogroup II. The description in lines 329-332 highlights only that the African G3 strains were G3P[8] or G3P[6] and were not similar to reports from other regions of G3 strains with differing P-types. This is noteworthy.
To the last comment (#3), we agree with the reviewer that this is a critical area of research. The African Rotavirus Surveillance Network is diligently working on this specific topic including research on the full genomic characterization of rotavirus strains across the continent, the relationship between the modest efficacy observed in Africa with various factors including host related factors (eg. HGBA and microbiome) and virus related factors. One reason why Africa is so active in clinical evaluation of new and novel rotavirus vaccine candidates, such as the RV3-BB vaccine, is because it has a different VP4 P[6] which may be relevant for African populations.
We will continue to conduct this research and continue to report it in scientific publications and in scientific meetings.
Reviewer 4 Report
The article is a detailed compendium of the topics presented during the 13th African Rotavirus 3 Symposium, held as a virtual event on 3rd and 4th No- 98vember 2021 due to COVID-19 pandemics.
The authors provided a glimpse of the rotavirus vaccination situation in African continent, mentioning among other topics the good situation of anti-rotavirus vaccination before pandemics, where 42 of the 54 countries fully implemented rotavirus immunization at the time of the last meeting in November 2019. Initially, the Symposium was thought to be presential in Kenya, a country that reduced in approximately 67% of rotavirus hospitalizations since vaccine introduction in 2012.
In this regard, authors remarked that Africa contributes approximately 80% of the global burden of rotavirus mortality (i.e., Nigeria and DRC have some of the highest rates of rotavirus mortality per 1,000 live births) so the scale-up of rotavirus vaccination across the continent would dramatically lessen the burden of rotavirus disease.
The authors included data about outstanding topics, as mentioned below:
Rotavirus Immunization in Africa and the impact of COVID-19: the introduction of vaccination in some countries, like Central African Republic and Nigeria, the switching of vaccines, and disruptions in health services (including vaccinations) were among the aspects affected by the pandemics.
Programmatic use of rotavirus vaccines in Africa: The need for vaccine product-specific surveillance for: progressive introduction of vaccination in some countries, intussusception risk monitoring, and challenges associated with the switch of products were highlighted.
Diarrhea and rotavirus surveillance in Africa
Authors described the data provided during the Symposium about the estimates of the etiology-specific burden of hospitalized diarrhea cases from the WHO Global Pediatric Diarrhea Surveillance Network (GPDS), which enrolled 33 countries from Africa, Asia, Latin America, and Europe in 2017 and 2018. Conclusions remarked that improving the efficacy and coverage of rotavirus vaccination and prioritizing interventions against other enteric pathogens including Shigella and norovirus could further reduce diarrhea morbidity and mortality.
Genomic analysis of rotavirus strains in Africa
Genomic data about Rotavirus circulating before and after vaccine introduction from Malawi, Ghana, Ethiopia, and South Africa, showed changing patterns of predominant genotypes between both periods, and antigenic changes within common human genotypes were observed. Authors remarked that the speaker’s conclusions about whether massive vaccination induced selective pressures favoring strains or constellations unusual in humans, and if such strains with animal rotavirus-like characteristics, or those with various amino acid changes on their outer capsid proteins, would have a negative impact on the vaccine effectiveness remain to be determined
Finally, authors described the perspective for Rotavirus vaccine research in Africa, and remarks on this topic highlighted that African researchers continue to lead the way with respect to the clinical evaluation of new and novel rotavirus vaccines. Studies about safety and immunogenicity of formulations of Vaccines made in India, were consistent with those observed in their licensure trials and support the use of India´s vaccines across wider geographical areas.
I consider that the article provided remarkable data about Rotavirus vaccination in Africa, which is of importance not only for the African continent but for global improvement in Rotavirus immunization programs, and doesn´t need any change.
Author Response
Thank you for your review of our manuscript.
As you've indicated, the meeting report presented here covers a wide range of topics on rotavirus burden in Africa, the epidemiology and genomic characterization of strains in circulation and on the impact of rotavirus vaccine use.
This manuscript provides an overview of rotavirus epidemiology and rotavirus vaccine use in Africa.